# Assessment of Foods Associated with Sodium and Potassium Intake in Japanese Youths Using the Brief-Type Self-Administered Diet History Questionnaire

**DOI:** 10.3390/nu13072345

**Published:** 2021-07-09

**Authors:** Masayuki Okuda, Satoshi Sasaki

**Affiliations:** 1Graduate School of Sciences and Technology for Innovation, Yamaguchi University, 1-1-1 Minami-Kogushi, Ube 755-8505, Japan; 2Department of Social and Preventive Epidemiology, School of Public Health, The University of Tokyo, 7-3-1 Hongo, Bunkyo-ku, Tokyo 113-0033, Japan; stssasak@m.u-tokyo.ac.jp

**Keywords:** adolescents, brief-type self-administered diet history questionnaire, children, foods, potassium, sodium, sodium-to-potassium ratio, urinary excretion

## Abstract

The identification of sodium and potassium intake in youths is an important step to preventing the increase of blood pressure in childhood. We examined food intake and estimated mineral intake using a brief-type self-administered diet history questionnaire (BDHQ) to test its validity as a comparison with urinary excretion in Japanese youths. The subjects were 5th and 8th graders (*n* = 2377), who completed the BDHQ and permitted the use of their overnight urine specimens. Sodium intake was poorly associated with sodium excretion (*Rho* = 0.048), and the coefficients of dietary potassium and a sodium-to-potassium molar ratio were 0.091–0.130. Higher soybean paste (*miso*) intake and pickles were significantly associated with higher sodium excretion (*p* ≤ 0.005). However, these foods were positively associated with potassium excretion (*p* = 0.002–0.012), and not associated with an excreted sodium-to-potassium ratio. Fruits and dairy products were positively associated (*p* ≤ 0.048), whereas beverages were negatively associated with potassium excretion (*p* ≤ 0.004). The association of the sodium-to-potassium ratio was opposite to that of potassium (*p* ≤ 0.001). The choice of foods, potassium, and the sodium-to-potassium ratio assessed using the BDHQ are available as part of health education for youths, but the assessment of sodium intake in population levels should be carefully conducted.

## 1. Introduction

High sodium intake is one of the major health risks in people worldwide. It contributes largely to mortality, especially in East Asia and West-Pacific areas, through cardiovascular diseases, cancers, and kidney diseases [1]. Even in children, high sodium intake is related to high blood pressure [2,3,4,5], and further meta-analyses showed tracking of higher blood pressure from childhood to adulthood [6,7]. The reduction of sodium intake is important to control blood pressure (a public health challenge) in children [8]. High potassium intake and a low dietary sodium-to-potassium ratio should be also considered, with measures against high sodium intake, to prevent the increase of blood pressure in childhood [9,10]. Despite scarce literature about the associations between high potassium intake or a low dietary sodium-to-potassium ratio and blood pressure in youths [11,12], these nutritional indices have beneficial possibilities on blood pressure [13].

The identification of mineral intake is the first step for policy planning [14]. Processed foods are the main sources of dietary sodium in the UK and US [15], where food reformulation, such as bakery products, processed meats, dairy products, sauces, and convenience meals, may be targeted [14,16]. Conversely, discretionary use during cooking or eating is the main contributor to sodium intake in East Asian countries, such as from soy sauce in Japan and salt in China [15,17]. Public health education or consumer awareness campaigns may be efficient for implementing sodium reduction interventions in these countries [14]. Most data about food sources that policymaking is based on is derived from adult surveys, but foods contributing to sodium intake vary among different age groups [17,18,19].

Understanding individual and people’s sodium intake is helpful for public health policy communication, raising motivation for habit change, and maintaining a healthy dietary habit. A brief and simple questionnaire may be useful for community settings. Uechi et al. reported that simple questions about consumption of soybean paste (*miso*) soup, salty foods, or noodle soup, and seasonings/condiments use were related to excess sodium intake measured using 24 h urinary sodium excretion in Japanese adults [20]. The score of the Salt Check Sheet consisting of 13 questions was correlated with 24-h urinary sodium excretion in adults [21]. The brief-type self-administered diet history questionnaire (BDHQ) with 64 food items, which takes 15–20 min to accomplish, is applicable for a large population survey [22]. Estimates from this questionnaire were associated with not only urinary sodium but also potassium excretion in adults [20,23]. The BDHQ for youths, with 55–64 food items modified from the adult version, may be useful to estimate sodium and potassium intake and foods related to high sodium and low potassium intake of children and adolescents. However, correlations between estimates from the BDHQ for youths and biomarkers other than sodium and potassium intake (nitrogen, carotenoids, tocopherols, and marine *ω*-3 polyunsaturated fatty acids) were slightly lower than those in adults [24,25]. This study investigated whether estimates of sodium, potassium, and foods from the BDHQ for youths were associated with urinary excretion to test its validity and explored food intake related to sodium and potassium intake. The results are expected to provide the possibilities and limitations of the BDHQ for youths as a tool in sodium- and potassium-related healthy diet education in school settings, since other simple questionnaires have not been validated with urinary excretion of youths.

## 2. Materials and Methods

### 2.1. Subjects

The subjects were 5th and 8th graders (aged 10–11 years and 13–14 years, respectively) at primary and junior high schools in Shunan City, Japan, in 2011. This study was a part of the Shunan Child Cohort study 2006–2012 described in other studies [26,27]. The subjects involved in this study participated in the questionnaire survey and urine sampling between April and May 2011.

### 2.2. Dietary Assessment

The BDHQ is a 4-page, self-administered questionnaire to assess dietary intake in the previous month, which was developed to estimate nutrients using the Japanese food composition table, 5th edition. The correlation coefficients of intake estimated between single-administered BDHQ and 16-d dietary records in adults were 0.44–0.51 for sodium, 0.56–0.64 for potassium, and 0.29–0.63 for foods (cereals, noodles, bread, pulses, confectionaries, vegetables, pickled vegetables, non-alcoholic beverages, soft drinks, fish, meat, and dairy products) [22]. For the subjects who did not drink alcoholic beverages, we used a BDHQ10y version (55 food items) in the 5th grader survey and a BDHQ15y version (64 items) in the 8th grader survey. In external validation with biomarkers, the Spearman correlation coefficients of corresponding nutrients intake using the BDHQ10y and BDHQ15y were 0.110–0.207 and 0.263–0.306 for serum carotenoids, respectively; 0.112–0.229 and 0.222–0.477 for red blood corpuscle marine *ω*-3 polyunsaturated fatty acids, respectively [24]. The correlation coefficient of protein intake with urinary nitrogen excretion in 7–9th graders was 0.109–0.302 [25]. The subjects completed the BDHQ at home once during the survey period. We used the data with plausible energy intake; ≥ 0.5 age- and sex-specific estimated energy requirements for low physical activity levels and ≤ 1.5 energy requirements for high physical activity levels [28]. Sodium and potassium are expressed as mg/day (*Na_BDHQ_*; mg = 23 × mmol, and *K_BDHQ_*; mg = 35.5 × mmol, respectively). The intake of sodium, potassium, and foods (Table 1) was energy-adjusted using an energy density method (/1000 kcal). The sodium-to-potassium ratio was calculated as a molar ratio of daily intake (*Na/K_BDHQ_* ratio).

### 2.3. Urinalysis

A first-void urine specimen after waking up (overnight urine) at home was collected for annual health checkups according to the School Health and Safety Act. We used this single collection for each subject. After a mandatory dipstick test, sodium (mmol/L) and potassium (mmol/L), and creatinine (mg/dL) concentrations in the remaining urine specimen were measured using the electrode method and enzyme method, respectively, at the Tokuyama Medical Association Hospital. Daily sodium or potassium excretion was estimated based on the ratio of sodium or potassium to creatinine concentrations in overnight urine multiplied by estimated daily creatinine excretion [29]. Comparing with measurements in the 24-h urine, the intraclass correlation coefficient and the mean estimate were 0.61 and −8.6% for sodium in 5–9th graders, respectively, and 0.55 and −8.6% for potassium, respectively. Urinary excretion of sodium and potassium was considered as 86% and 77% of intake, respectively [30]. Finally, daily sodium, and potassium intake were expressed as mg/day (*Na_ex_*, and *K_ex_*, respectively). *Na_ex_* and *K_ex_* divided by body weight (mg/day·kg) [31] were also used as an alternative intake criterion. The sodium-to-potassium ratio was also calculated as a molar ratio of daily excretion (*Na/K_ex_* ratio).

### 2.4. Confounders

Body height (cm) and weight (kg) were put in the BDHQ by the respondents. Body mass index (BMI; kg/m^2^) was calculated as weight/square of height × 10^−4^. Self-reported BMI before the anthropometrics at the health checkups was highly correlated with measured BMI (Pearson correlation coefficients = 0.930–0.964), and the mean difference between them was −0.224 to 0.101 kg/m^2^ in the 2006–2010 surveys [32]. A z-score BMI (zBMI) was obtained using the lambda-mu-sigma method [33], based on the 2000 Japanese reference [34]. We used the data with BMI correspondent ≥ 17 kg/m^2^ or ≤ 30 kg/m^2^ at 18 years for analysis to exclude possible misentry of height and weight. Ages were calculated as the difference between the birth date and the date when completing the questionnaire divided by 365.25.

### 2.5. Statistical Analysis

Subjects for analysis were selected as seen in Figure 1. Variables are presented as mean ± standard deviation. The Spearman correlation (*Rho*) was used to examine the associations between *Na_BDHQ_, K_BDHQ_*, or the *Na/K_BDHQ_* ratio and corresponding *Na_ex_*, *K_ex_*, or the *Na/K_ex_* ratio, respectively. The subjects were stratified into five groups based on the grade- and sex-specific quintiles of each food group. The least-square means of *Na_BDHQ_, K_BDHQ_*, and the *Na/K_BDHQ_* ratio in the quintile strata were adjusted for sex, age, zBMI, and intake energy using analysis of covariance (ANCOVA), and ordinal linear trends across the quintile strata were tested to explore which food intake was associated with mineral intake. SAS version 9.4 (SAS inc. USA) was used for data analysis, where a *p*-value < 0.05 was considered significant.

## 3. Results

Demographics, intake, and urinary excretion of minerals are presented in Table 2. The ages of the 5th and 8th graders were 10.56 ± 0.29 years and 13.59 ± 0.29 years, respectively. *Na_BDHQ_*, *K_BDHQ_*, and the *Na/K_BDHQ_* ratio estimated from BDHQ were 4179 ± 1162 mg/day (10.6 ± 3.0 salt-equivalent g/day), 2491 ± 777 mg/day, and 3.4 ± 1.7, respectively, and *Na_ex_*, *K_ex_*, and the *Na/K_ex_* ratio estimated from overnight urine were 3394 ± 2072 mg/day (8.6 ± 5.3 salt-equivalent g/day), 1759 ± 1275 mg/day, and 2.7 ± 0.6, respectively.

The correlation coefficient between *Na_BDHQ_* and *Na_ex_* was 0.048 (*p* = 0.031; Table 3). Using energy-adjusting *Na_BDHQ_*, or *Na_ex_*/weight did not improve the correlation coefficients but attenuated them, and the associations lost significance (*Rho* = 0.036–0.044). The correlation coefficients for potassium and sodium-to-potassium ratio were 0.091 and 0.096, respectively. When energy-adjusted *K_BDHQ_* and weight-adjusted *K_ex_* were used, the coefficients improved to 0.108–0.130.

The association between food intake classified based on quintiles of intake and urinary excretion of minerals was examined using ANCOVA (Table 4, and Appendix A). Regarding *miso*, the highest intake group (Q5) excreted more *Na_ex_* than the lowest group (Q1) did by 495 mg/day, and the trend was significant (*p_trend_* < 0.001). The association for pickles was also significant (*p_trend_* = 0.005; the difference between Q1 and Q5, 319 mg/day). Other foods were not significantly associated with *Na_ex_*. High intake of *miso* and pickles was significantly and positively associated with *K_ex_* as with *Na_ex_* (*p* = 0.002 and 0.012, respectively), whereas these foods were not associated with the *Na/K_ex_* ratio. In contrast, fruits and dairy products had favorable effects; their high intake was associated with high *K_ex_* and a low *Na/K_ex_* ratio (*p* ≤ 0.048). Beverages, especially sugar-sweetened beverages, were significantly associated with low *K_ex_* (*p* = 0.004, and 0.024, respectively), and beverages, and seasonings/condiments had unfavorable effects on the *Na/K_ex_* ratio (*p* < 0.001, and = 0.039, respectively). The negative association between confectionaries and *K_ex_*, and the positive association between noodles and the *Na/K_ex_* ratio were marginally significant, but other foods were not significantly associated with *K_ex_* or the *Na/K_ex_* ratio.

## 4. Discussion

The mean *Na_ex_* estimated using the overnight urine of the 5th and 8th graders was 3394 ± 2072 mg/day. *Na_BDHQ_* was 4179 ± 1162 mg/day, which was not only overestimated, but barely correlated with *Na_ex_* estimated from overnight urine (*Rho* < 0.1). The mean *K_ex_* was 1759 ± 1275 mg/day, whereas *K_BDHQ_* was 2491 ± 777 mg/day, which was overestimated, but their association was significant. The correlation of the sodium-to-potassium ratio between the BDHQ and urinary excretion was also significant. *Miso* and pickles estimated from the BDHQ were significantly associated with *Na_ex_* and *K_ex_*, but not with the *Na/K_ex_* ratio. The intake of fruits and dairy products was favorable because it was associated with the high *K_ex_* and low *Na/K_ex_* ratio, and the intake of beverages, sugar sweetened beverages, and seasonings was unfavorable.

### 4.1. Foods Associated with Sodium, Potassium, and Sodium-to-Potassium Ratio

Several foods were positively or negatively related to *Na_ex_*, *K_ex_*, or the *Na/K_ex_* ratio, but there were neither foods with common beneficial effects on *Na_ex_*, *K_ex_*, the *Na/K_ex_* ratio, nor foods with common adverse effects. For example, the intake of *miso* and pickles was significantly associated with high *Na_ex_*, and positively associated with high *K_ex_*. These foods are blamed for high sodium intake in the Japanese diet, but rather *miso* had a beneficial effect on blood pressure [35,36]. In studies concurrently assessing sodium, potassium, and a sodium-to-potassium ratio of youths, sodium assessed from urinary excretion or dietary records was not associated with blood pressure, but potassium was negatively associated, and the sodium-to-potassium ratio was positively associated with it [11,12]. Similar findings were seen in studies on adults using a 24-h urine collection and dietary assessment [37]. When focusing on controlling the population’s blood pressure, dietary habits and patterns with foods lowering the sodium-to-potassium ratio, high proportions of fruits and dairy products, and low proportions of beverages and seasonings/condiments should be recommended. 

It is difficult to explain why beverages were related to low potassium intake and a high dietary sodium-to-potassium ratio. In a study in the United Kingdom, people with a high consumption of ultra-processed foods had a higher intake of sodium and lower intake of potassium [38]. Japanese adolescents who frequently consumed take-out dishes and food from convenience stores had higher urinary sodium excretion than adolescents who consumed these foods less frequently [39]. Consumption of beverages may be related to increased consumption of processed foods and out-of-home behaviors.

### 4.2. Sodium Intake

In this study, *miso* and pickles were found to have a significant association with *Na_ex_*. This assessment is available for food education to reduce sodium intake, but the assessment of major sources, such as seasonings/condiments, cereals, and fish, is not very useful to elucidate high sodium intake. In a previous study of Japanese adults, where a dietary assessment was designed to measure discretionary salt intake with scaling discretionary amounts of soy sauce, the major contributors to total sodium intake were significantly associated with urinary sodium excretion [31]. The BDHQ was designed to capture Japanese dietary habits, but is not specific for sodium intake.

### 4.3. Potassium Intake

*K_BDHQ_* was overestimated more than *Na_BDHQ_* was with excretion as references. Sodium and potassium are excreted in sweat and feces in addition to urine. When calculating *Na_ex_* and *K_ex_*, we considered this extra-urinary excretion larger for potassium than for sodium. Extra-urinary excretion could be underestimated. Loss of potassium during cooking [40,41,42] is another explanation for the overestimation of *K_BDHQ_*. Unlike sodium from seasonings/condiments, potassium is rich in various core foods, such as vegetables, fish, and meats [43]; skin-stripping and boiling can lead to the loss of potassium during cooking.

In contrast, the association between intake and excretion was stronger for potassium than for sodium. It is difficult for youths to know discretionary use of sodium during cooking and at the table, unlike the intake of core foods contributing to potassium intake. The dietary habits of youths with regard to high intake of potassium, rather than sodium, can be determined from the BDHQ assessment. However, we did not find a positive association of vegetables and fish as seen in a study of Japanese adults [44]. Other than the explanations mentioned above, various foods included in food groups and knowledge about favorable effects of vegetables and fish may attenuate or obscure the association between the intake of core foods and *K_ex_*.

### 4.4. Simple Questionnaires

Typical food frequency questionnaires have 80 to 120 food items. The BDHQ, which was simplified from the DHQ with 151 food items to reduce a respondent burden [22,45], is available to enhance self-awareness and assess population levels of dietary intake in school settings. In addition to the BDHQ, simple Japanese questionnaires have been used [46,47,48]. The Salt Check Sheet has 13 items and a score range of 0–35. Correlation coefficients of the scores for adults were 0.30 with estimates from spot urine [47], and 0.27 with measurements from 24-h urine [21]. Its validity has not been examined using urinary sodium excretion in youths, but a correlation coefficient between the scores and *Na_BDHQ_* was 0.408 for 5th and 7th graders [49], which was higher than those with urinary estimates for adults (0.27–0.30). The high correlation between the Salt Check Sheet and the BDHQ may be explained by similar items used for estimation. There were other brief questionnaires for estimation of salt intake, originating from a tool of the Japanese Ministry of Health and Welfare [46,50,51]. One was used for adolescents and revealed the association between salt intake and blood pressure [51]. However, validation was examined comparing with weighted dietary records, which showed a correlation coefficient of 0.28 in adults. If the coefficients with urinary sodium would be used, the correlations may be lower. Food frequency questionnaires reduced the burden of target subjects, added to which the BDHQ can assess dietary potassium and a sodium-to-potassium ratio other than dietary sodium.

### 4.5. Meeting the References

A proposed tentative dietary goal for preventing lifestyle diseases (DG) was to reduce salt intake in Japan: <6.0 g/day for both sexes aged 10–11 years, and <6.5 and <7.0 g/day for girls and boys aged 12–14 years, respectively [28]. The DG for potassium is ≥2000 and ≥2200 mg/day for girls and boys aged 10–11 years, respectively, and 2400 mg/day for both sexes aged 12–14 years. The mean salt-equivalent *Na_ex_* of the subjects in this study was higher, and *K_ex_* was lower than the DGs for corresponding ages, but several considerations were needed before comparing them. First, a gold standard for estimating daily sodium and potassium intake is measuring 24-h urinary excretion. The equation to estimate *Na_ex_* and *K_ex_* from overnight urine was developed using 24-h urinary excretion with an intraclass correlation (validity coefficient) of 0.55–0.61 [29]. Second, daily sodium and potassium intake vary each day. This intra-individual variation is quantifiable by repeated measurement, but only single overnight urine specimens were available in this study. These measurement errors inflate the standard deviation of *Na_ex_* and *K_ex_* and attenuate correlation coefficients with other variables, such as the estimates from the BDHQ [52,53]. At least more than half of the youths may overconsume sodium, and another more than half may have insufficient potassium intake because the mean salt-equivalnet *Na_ex_* was higher, and the mean *K_ex_* was lower than the DGs. Moreover, the sodium-to-potassium ratio in this population highly exceeded the recommendation of the World Health Organization (1.0) [8]. 

### 4.6. Strengths and Limitations

A high retention rate and a large sample size were the strengths of this study. On the other hand, several limitations should be considered to interpret the results. First, it is possible that the foods associated with sodium, potassium, and the sodium-to-potassium ratio in this study indirectly link to the intake of these minerals. Collateral food intake could influence the associations, but youths can evaluate their dietary intake from the BDHQ, which would help them modify their dietary habits. In other words, a dietary pattern or quality assessment have the potential to elucidate the association with the health outcomes. Second, correlations in this study were attenuated because of the single overnight urine specimen, as mentioned above. Repeated urinary excretion measurement would be better. Third, the calculation of *Na_ex_* and *K_ex_* was based on self-reported anthropometrics. We excluded the subjects with obesity, whose daily creatinine excretion would be overestimated [29]. Fourth, the survey was restricted to a local city, and the results have limited generalizability to other populations. Possible regional variability of dietary habits in Japan should be elucidated in the future.

## 5. Conclusions

*Na_BDHQ_* estimated using the BDHQ for youths was associated with urinary sodium excretion, but the correlation coefficient was lower than 0.1. Instead, *K_BDHQ_* and the *Na/K_BDHQ_* ratio had higher correlations with urinary excretion than *Na_BDHQ_* did in Japanese youths. *Miso* and pickles intake measured using the BDHQ for youths was associated with *Na_ex_*; the assessment of foods instead of sodium intake is better to estimate sodium consumption as alternatives and allow youths to evaluate their own dietary habits. Considering the importance of prevention of hypertension from childhood, dietary habits with high proportions of fruits and dairy products, and avoidance of high intakes of beverages, seasonings/condiments are encouraged to reduce the dietary sodium-to-potassium ratio. After deciding the target nutrients, the BDHQ is available as a school health education tool. When using the BDHQ to assess youth sodium and potassium intake, overestimation, a sample size, and possible confounders should be taken account of.

## Figures and Tables

**Figure 1 nutrients-13-02345-f001:**
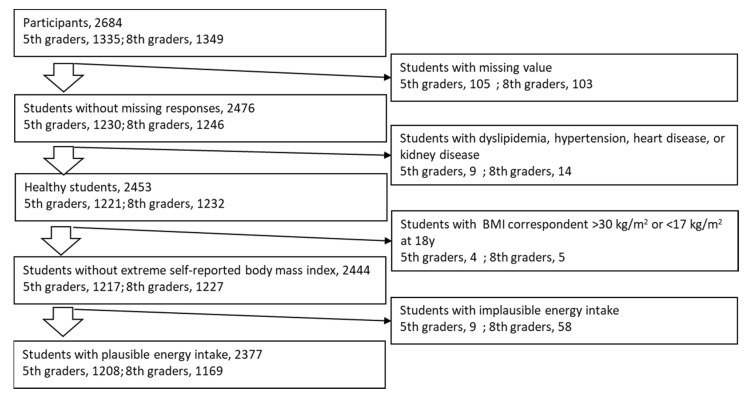
Subject selection.

**Table 1 nutrients-13-02345-t001:** Food groups.

Food Groups	Food Items
Meat	Poultry, meat, processed meat, liver
Fish	Squid and octopus, shellfish, fish with bone, tuna, oily fish, dried fish, lean fish, fish paste
Vegetables	Raw vegetables used in salad, green leafy vegetables, broccoli, cabbage, Chinese cabbage, carrots, pumpkins, radishes and turnips, other root vegetables (onions, burdock, lotus root), tomato sauce, boiled tomato and stewed tomato, salted vegetable pickles
Fruits	Citrus fruits, strawberries, persimmons, kiwi fruit, other fruits
Cereals	Bread: including white bread and sweet Japanese bread Noodles: buckwheat noodles, Japanese wheat noodles; instant noodles, Chinese noodles, spaghetti, macaroni
Seasonings/condiments	Butter, margarine, jams, mayonnaise, ketchup, soy, and other sauces
Fermented soybean paste (*miso*)	
Noodle soup	
Confectionaries	Cookies, biscuits, Japanese sweets, rice crackers, rice cakes, Japanese-style pancakes, snack confectionaries
Beverages	Green tea, black tea, oolong tea, coffee, fruits and vegetable juice, sugar-sweetened beverages
Sugar-sweetened beverages	Cola, sweetened soft drink, coffee with milk, lactobacillus beverage
Dairy products	Milk, low-fat milk, yogurt, cheese

**Table 2 nutrients-13-02345-t002:** Characteristics of the subjects.

	*n* = 2377
Age, years	12.05 ± 1.54
Male: female	1207: 1170
Height, cm	147.4 ± 11.2
Weight, kg	39.9 ± 9.9
Body mass index, kg/m^2^	18.10 ± 2.59
zBMI	0.02 ± 1.03
BDHQ	
Energy	2005 ± 570
*Na_BDHQ_*, mg/day	4179 ± 1162
*Na_BDHQ_*, mg/day·1000 kcal	2123 ± 402
*K**_BDHQ_*, mg/day	2491 ± 777
*K_BDHQ_*, mg/day·1000 kcal	1260 ± 263
*Na/K**_BDHQ_* ratio, mol/mol	3.4 ± 1.7
Urinary Excretion	
*Na_ex_*, mg/day	3394 ± 2071
*Na_ex_*/weight, mg/day·kg	86.6 ± 4702
*K_ex_*, mg/day	1759 ± 1275
*K_ex_*/weight, mg/day·kg	44.9 ± 31.2
*Na/K_ex_* ratio, mol/mol	2.7 ± 0.6

zBMI, z-score body mass index; BDHQ, brief-type self-administered diet history questionnaire; *Na_BDHQ_*, *K_BDHQ_*, and *Na/K_BDHQ_* ratio: sodium, potassium intake, and dietary sodium-to-potassium molar ratio estimated from the BDHQ; *Na_ex_*, *K_ex_*, and *Na/K_ex_* ratio: sodium intake, potassium intake, and dietary sodium-to-potassium molar ratio estimated from overnight urine.

**Table 3 nutrients-13-02345-t003:** Spearman’s correlation coefficients (*Rho*) between BDHQ estimation and corresponding urinary excretion (*n* = 2377).

	Urinary Excretion			
	Raw Values, mg/day		Weight Adjusted Values, mg/day·kg	
	*Rho*	*p*	*Rho*	*p*
*Na**_BDHQ_*, mg/day	0.048	0.031	0.037	0.074
*Na**_BDHQ_*, mg/day·1000 kcal	0.044	0.081	0.036	0.081
*K_BDHQ_*, mg/day	0.091	<0.001	0.108	<0.001
*K_BDHQ_*, mg/day·1000 kcal	0.110	<0.001	0.130	<0.001
*Na/K**_BDHQ_**ratio,* mol/mol	0.096	<0.001	0.130	<0.001

Coefficients were adjusted for sex, age, intake energy, and zBMI. BDHQ, brief-type self-administered diet history questionnaire; zBMI, z-score body mass index; *Na_BDHQ_*, *K_BDHQ_*, and *Na/K_BDHQ_* ratio: sodium, potassium intake, and dietary sodium-to-potassium molar ratio estimated from the BDHQ.

**Table 4 nutrients-13-02345-t004:** Least square means of minerals estimated from overnight urine (g/day) across quintile strata of food intake.

	*Na_ex_* (mg/day)						*K_ex_* (mg/day)						*Na/K_ex_* Ratio					
	Q1	Q2	Q3	Q4	Q5	*p_trend_*	Q1	Q2	Q3	Q4	Q5	*p_trend_*	Q1	Q2	Q3	Q4	Q5	*p_trend_*
Cereals	3376	3451	3463	3371	3576	0.289	1763	1832	1734	1684	1784	0.573	3.3	3.3	3.4	3.4	3.5	0.065
Bread	3447	3442	3495	3371	3481	0.990	1870	1686	1670	1806	1765	0.618	3.2	3.5	3.5	3.4	3.3	0.762
Noodles	3358	3440	3495	3406	3538	0.272	1743	1727	1819	1777	1730	0.898	3.3	3.4	3.4	3.3	3.6	0.070
Vegetables	3375	3352	3577	3531	3400	0.438	1777	1692	1748	1798	1782	0.517	3.4	3.4	3.5	3.4	3.3	0.308
Pickles	3281	3405	3411	3579	3600	0.005	1694	1680	1732	1855	1831	0.012	3.3	3.4	3.4	3.4	3.5	0.129
Fruits	3408	3608	3403	3490	3327	0.349	1695	1726	1759	1759	1858	0.048	3.5	3.6	3.3	3.4	3.1	<0.001
Fish	3501	3437	3444	3434	3421	0.581	1727	1779	1770	1812	1708	0.973	3.5	3.4	3.4	3.3	3.4	0.266
Salty fish	3455	3599	3399	3409	3375	0.238	1779	1714	1804	1757	1742	0.861	3.3	3.6	3.3	3.4	3.3	0.184
Meat	3488	3477	3360	3490	3421	0.685	1754	1740	1803	1772	1727	0.906	3.4	3.4	3.3	3.4	3.4	0.722
Processed meat	3378	3536	3470	3362	3490	0.864	1738	1812	1715	1758	1772	0.939	3.4	3.4	3.5	3.3	3.4	0.875
Dairy product	3436	3528	3429	3376	3467	0.279	1608	1768	1690	1799	1932	<0.001	3.6	3.5	3.4	3.3	3.1	<0.001
Confectionaries	3518	3604	3315	3376	3423	0.162	1781	1856	1785	1687	1688	0.053	3.4	3.3	3.3	3.5	3.5	0.226
Japanese conf.	3576	3471	3419	3461	3307	0.064	1774	1745	1791	1784	1701	0.558	3.4	3.5	3.4	3.3	3.3	0.213
Beverages	3423	3436	3397	3498	3482	0.553	1910	1783	1710	1721	1673	0.004	3.1	3.3	3.4	3.5	3.6	<0.001
SSBs	3540	3427	3328	3461	3481	0.775	1882	1687	1826	1764	1639	0.024	3.3	3.5	3.3	3.3	3.5	0.218
Seasonings/condiments	3419	3456	3455	3526	3379	0.977	1767	1801	1775	1751	1702	0.326	3.3	3.4	3.4	3.4	3.5	0.039
Soy and other sauces	3343	3386	3609	3495	3402	0.604	1735	1751	1827	1706	1777	0.828	3.3	3.4	3.4	3.3	3.6	0.099
Noodle soup	3362	3506	3428	3335	3606	0.287	1728	1830	1807	1731	1701	0.567	3.3	3.2	3.5	3.5	3.5	0.060
Soybean paste (*miso*)	3238	3358	3436	3473	3733	<0.001	1650	1693	1791	1760	1903	0.002	3.4	3.4	3.3	3.4	3.4	0.945

The least-square means were adjusted with sex, a z-score body mass index, and energy intake. Japanese conf.: Japanese confectionaries (Japanese sweets, rice crackers, rice cakes, and Japanese-style pancakes); SSBs: sugar-sweetened beverages; *Na_ex_*, *K_ex_*, and the *Na/K_ex_* ratio: sodium intake, potassium intake, and the dietary sodium-to-potassium molar ratio estimated from overnight urine.

## Data Availability

The data presented in this study are available on request from the corresponding author.

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
