# Peer review of "Assessment of Foods Associated with Sodium and Potassium Intake in Japanese Youths Using the Brief-Type Self-Administered Diet History Questionnaire"

_nutrients, 2021, doi:10.3390/nu13072345_

Round 1

Reviewer 1 Report

This study examined the validity of using a self-administered diet history questionnaire to estimate the salt and potassium intake in Japanese youth. This topic is exciting and closely relevant to nutrition in youth. A few comments below:

  1. In the abstract, the authors should mention that one of the major goals of this study was to test the validity of BHDQ in youth.
  2. First sentence of the introduction, is it more appropriate to say "high sodium" intake is one of the major health risks...
  3. Methods, 2.5: It is unclear what five groups the study population was divided into. In addition, it seems like the majority of analyses were conducted in the total study population.
  4. Since the correlations were examined only in this cohort, these findings may have limited generalizability to other populations. It would be great if the authors could comment on this limitation and the public health implications about how these findings can be used in epidemiologic studies examining associations between salt intake and health outcomes in children and youth.
  5. The authors conclude that assessing food items like Miso and pickles instead of salt intake from FFQ is better to estimate salt consumption in children and youths. In this way, is it possible to calculate a more accurate salt intake based on this food that can be used in public health prevention and epidemiologic studies?

Author Response

Response to Reviewer 1

Dear Reviewer 1,

We are thankful for your positive comments and helpful advice. We have corrected and revised the manuscript as the followings. We have marked the corrections in red in the revised manuscript. We hope that this may improve our manuscript.

Sincerely,

Masayuki Okuda

  1. In the abstract, the authors should mention that one of the major goals of this study was to test the validity of BHDQ in youth.

 Response 1:  We have mentioned “testing the validity” in the Abstract and Introduction.

“We examined food intake and estimated mineral intake using a brief-type self-administered diet history questionnaire (BDHQ) to test its validity as comparison with urinary excretion in Japanese youths.” Line 13

“This study investigated whether estimates of sodium, potassium, and foods from the BDHQ for youths were associated with urinary excretion to test its validity” Line 69

  1. First sentence of the introduction, is it more appropriate to say "high sodium" intake is one of the major health risks...

Response 2: We have corrected it.

“High sodium intake is one of the major health risks in people worldwide.” Line 30

  1. Methods, 2.5: It is unclear what five groups the study population was divided into. In addition, it seems like the majority of analyses were conducted in the total study population.

 Response 3: We have changed categorization to stratification as the followings.

“The subjects were stratified into five groups based on the grade- and sex-specific quintiles of each food group. The least-square means of NaBDHQ, KBDHQ, and Na/KBDHQ in quintile strata were adjusted for sex, age, zBMI, and intake energy using analysis of covariance (ANCOVA), and ordinal linear trends across quintile strata were tested to explore which food intake was associated with sodium intake.” Lines 135–139 (137–142 in the All Markup).

The title of Table 4. “Least square means of minerals estimated from overnight urine (g/day) across quintile strata of food intake.”

The title and footnote of Supplementary Table S1. “Distribution of food intake as median of quintile strata (g/1000kcal).”, and “Stratification was based on grade- and sex-specific quintiles.”

  1. Since the correlations were examined only in this cohort, these findings may have limited generalizability to other populations. It would be great if the authors could comment on this limitation and the public health implications about how these findings can be used in epidemiologic studies examining associations between salt intake and health outcomes in children and youth.

Response 4: Thank you for your advice. We have added explanations about limited generalizability, and have added availability of a dietary pattern from the BDHQ.

“Fourth, the survey was restricted to a local city, and the results have limited generalizability to other populations.” Lines 300–302 (308–309 in the All Markup)

“In other words, a dietary pattern or quality assessment have the potential to elucidate the association with the health outcomes.” Lines 294–296 (302–304 in the All Markup)

  1. The authors conclude that assessing food items like Miso and pickles instead of salt intake from FFQ is better to estimate salt consumption in children and youths. In this way, is it possible to calculate a more accurate salt intake based on this food that can be used in public health prevention and epidemiologic studies?

Response 5: Thank you for your advice. It is a good idea. Calculating nutrients from the BDHQ responses is a buildup of nutrients of various foods based on food composition table, but is not estimation using regression models. Therefore, regression methods are out of the aim of this study. We have added the following explanation.

“The BDHQ is a 4-page self-administered questionnaire to assess dietary intake in the previous month, which was developed to estimate nutrients using the Japanese food composition table, 5th edition.” Lines 82–84

In addition, we have added the following sentence to utilize the BDHQ as a public health tool.

“In other words, a dietary pattern or quality assessment have the potential to elucidate the association with the health outcomes.” Lines 294–296 (302–304 in the All Markup)

Reviewer 2 Report

The authors propose the results of a cross-sectional study for the validation of a self-administered diet history questionnaire on a sample of 2377 young Japanese individuals belonging to two age groups: 10-11 years and 13-14 years. The purpose of the study is to evaluate the nutritional habits that emerged through the questionnaire to extrapolate the sodium and potassium intakes and evaluate the correlations between the intakes and levels of sodium and potassium or the ratio of sodium and potassium through overnight urine samples.

The study showed that some foods are more related to urinary excretion. For example, there is a significant correlation between pickles and miso paste with urine salt concentrations, while the correlation with potassium has been shown for foods such as pickles, fruits and dairy products, beverages and miso paste. Furthermore, the sodium-potassium ratio correlated significantly with the intake of fruits, dairy products, beverages, seasonings and condiments. As regards the correlation between intake and urinary excretion of individual minerals, salt shows a low correlation that lost significance after normalization for caloric intake and weight. While potassium intakes show a better correlation with excretion.

The manuscript is clear and fluent. It explored a questionnaire validation gap regarding sodium and potassium intake in children which is of great interest for the prevention of hypertension in children. The supporting bibliography is complete and well-focused on the treated topic.

Even if the sample used is conspicuous, some limits remain regarding the clinical relevance of the results obtained due to the limited significance of the correlations highlighted.

Here are some suggestions for revising the manuscript:

- Although with careful reading it can understand the nature of the coefficients to which the authors refer, in the manuscript, there is the risk of confusion between P, p and ρ. I recommend using only the lowercase letter p to indicate the p-value of statistical significance and use the term Rho or R to indicate the Spearman coefficient in the text and tables.

- I don't think it's correct to talk about urinary salt. It would be more correct to consider dietary sodium and urinary sodium when evaluating correlations. It would be more correct to convert the salt intake into sodium.

- In lines 152-156 the authors state that normalization of sodium excretion does not improve the correlation, but it would be more correct to state that normalization by weight and/or by caloric intake implies the loss of significance.

- It is not specified whether the samples for each individual were tested once or with replicates

Minor aspects:

- A repeated typo of the word "studnets" instead of "students" in figure 1

- A spacing after tables 2 and 3 would be useful

Author Response

Response to Reviewer 2

Dear Reviewer 2,

We are thankful for your positive comments and sound advice. We have corrected and revised the manuscript as the followings. We have marked the corrections in red in the revised manuscript. We hope that this may improve our manuscript.

Sincerely,

Masayuki Okuda

  1. Although with careful reading it can understand the nature of the coefficients to which the authors refer, in the manuscript, there is the risk of confusion between P, p and ρ. I recommend using only the lowercase letter p to indicate the p-value of statistical significance and use the term Rho or R to indicate the Spearman coefficient in the text and tables.

Response 1: Thank you for your advice. As per your recommendation, we have used the lowercase letter p for the probability value, and Rho for the Spearman coefficients in the Abstract, Results, and Discussion. Fonts in red.

  1. I don't think it's correct to talk about urinary salt. It would be more correct to consider dietary sodium and urinary sodium when evaluating correlations. It would be more correct to convert the salt intake into sodium.

Response 2: Thank you for your advice. We have changed “salt” to “sodium”, or “Na” throughout the manuscript, and calculated daily sodium intake (mg/day) in the Methods, Results, and Tables 2, 3, and 4. Fonts in red.

  1. In lines 152-156 the authors state that normalization of sodium excretion does not improve the correlation, but it would be more correct to state that normalization by weight and/or by caloric intake implies the loss of significance.

Response 3: We have changed this sentence.

“Using energy-adjusting NaBDHQ, or Naex/weight did not improve the correlation coefficients but attenuated them, and the associations lost significance (Rho = 0.036–0.044).” Lines 158–159 (163–165 in the All Markup)

  1. It is not specified whether the samples for each individual were tested once or with replicates

Response 4: We used one urine specimen and one response to the BDHQ for each subject. We have added the following sentences in the Methods.

“The subjects completed the BDHQ at home once during the survey period.” Lines 95–96

“We used this single collection for each subject.” Lines 106–107 (107–108 in the All Markup).

  1. A repeated typo of the word "studnets" instead of "students" in figure 1

Response 5: Thank you. We have corrected them and replaced Figure 1.

6.A spacing after tables 2 and 3 would be useful.

Response 6: We used “space before the paragraph” function of the WORD, but the Editor might delete it. This time we added vacant lines.

Round 2

Reviewer 2 Report

The authors improved the manuscript by revising it based on suggestions

Author Response

Thank you for your suggestions and recommendation.